# THE CASE FOR FULL-MATRIX ADAPTIVE REGULARIZATION

## ABSTRACT

Adaptive regularization methods pre-multiply a descent direction by a preconditioning matrix. Due to the large number of parameters of machine learning problems, full-matrix preconditioning methods are prohibitively expensive. We show how to modify full-matrix adaptive regularization in order to make it practical and effective. We also provide novel theoretical analysis for adaptive regularization in *non-convex* optimization settings. The core of our algorithm, termed GGT, consists of efficient inverse computation of square roots of low-rank matrices. Our preliminary experiments underscore improved convergence rate of GGT across a variety of synthetic tasks and standard deep learning benchmarks.

## 1 INTRODUCTION

Stochastic gradient descent is the workhorse behind the recent deep learning revolution. This simple and age-old algorithm has been supplemented with a variety of enhancements to improve its practical performance, and sometimes its theoretical guarantees.

Amongst the acceleration methods there are three main categories: momentum, adaptive regularization, and variance reduction. Momentum (in its various incarnations, like heavy-ball or Nesterov acceleration) is the oldest enhancement. It has a well-developed theory, and is known to improve practical convergence in a variety of tasks, small and large. It is also easy to implement. Variance reduction is the most recent advancement; in theory and practice, it is mostly applicable to convex optimization, and is thus less influential in deep learning.

This brings us to adaptive regularization: the most sophisticated, hard to implement, and debated acceleration method. While state-of-the-art optimizers such as Adam and AdaGrad (Kingma & Ba, 2014; Duchi et al., 2011) do use adaptive regularization, they do so in a very limited form: with diagonal matrices, often marketed as per-coordinate adaptive learning-rate methods. Despite solid theoretical guarantees, the practical value of diagonal adaptive regularization as compared to "vanilla" SGD has been the subject of much debate (Wilson et al., 2017). However, the efficacy of full-matrix adaptive regularization has been relatively unexplored. This is due to the prohibitive computational cost associated with full-matrix operations: full AdaGrad requires taking the inverse square root of a large matrix.

In this paper, we present GGT, a practical solution to the computational problems plaguing full-matrix adaptive regularization, making this technique scalable for modern deep models. At the heart of our method is a simple, GPU-friendly way to apply the inverse square root of the low-rank second-moment matrix of recent gradients; see Figure 1. GGT's running time is comparable to state-of-the-art optimizers.

We proceed to show that full-matrix preconditioning allows for much better exploitation of anisotropic curvature in loss landscapes. First, we show synthetic experiments which demonstate clear benefits of GGT over baselines, especially when the problem is ill-conditioned. Then, we implement GGT at scale, and show that the benefits translate to faster training on standard deep learning benchmarks. Our improvement is most salient in complicated landscapes like RNN training.

Our algorithm comes with theoretical guarantees. We give the first proof of convergence to first-order critical points for an algorithm with adaptive regularization in a stochastic non-convex setting, featuring a rate which is dependent on an *adaptive ratio*. We show examples where our bound is stronger than that for SGD, providing some theoretical basis for our empirical findings.

## 1.1 RELATED WORK

Since the introduction of AdaGrad (Duchi et al., 2011), diagonal adaptive regularization has been a mainstay in the machine learning practitioner's toolbox. A quick perusal of the literature shows that these methods have continued to thrive in the deep learning era, and appear in all major frameworks (Abadi et al., 2016; Paszke et al., 2017; Chen et al., 2015). By citation count (or GitHub search hits), Adam (Kingma & Ba, 2014) is by far the most popular adaptive optimizer for training a variety of modern deep models. For this reason, this paper's exposition is targeted towards a full-matrix drop-in replacement for Adam; however, our techniques extend straightforwardly to a plethora of variants, like RMSprop (Tieleman & Hinton, 2012), Adadelta (Zeiler, 2012), Nadam (Dozat, 2016), etc.

Full-matrix adaptive regularization has existed alongside the more commonly used diagonal-matrix manifestation since their common inception in (Duchi et al., 2011); however, a major obstacle to the scalability of these methods is the need for the storage and inversion of square matrices in the model dimension. This becomes prohibitively expensive in dimension greater than $10^4$, while state-of-the-art models regularly exceed $10^7$ parameters.

Matrix sketching has been employed to approximate the AdaGrad preconditioner (Krummenacher et al., 2016; Mehta et al., 2016); however, the sketched estimate for the matrix inverse can be sensitive to noise. In the former, the authors report a 5-10× overhead over AdaGrad, even with $< 10^5$ model parameters; we could not find a usable GPU implementation for their requisite rank-1 QR update. (Gupta et al., 2018) propose a way to do AdaGrad with Kronecker products of full-matrix preconditioners, a more limited setting which requires knowledge of the model's structure. Finally, as we argue in Section 3.1, there is intrinsic value of "forgetting" past curvature using an exponential window. With this, a low-rank preconditioning matrix naturally arises, allowing us to bypass the computational need for sketching in the model dimension or architecture-dependent restriction of the preconditioner.

Our algorithm bears a superficial resemblance to L-BFGS (Liu & Nocedal, 1989), a version of BFGS (Broyden, 1970; Fletcher, 1970; Goldfarb, 1970; Shanno, 1970) which uses a sliding window of gradient history. Although some are viable for large-scale implementation, these quasi-Newton methods, along with (subsampled, online, cubic-regularized) Newton methods (Erdogdu & Montanari, 2015; Agarwal et al., 2017b; Luo et al., 2016; Hazan et al., 2007; Agarwal et al., 2017a; Carmon et al., 2017) exhibit very different dynamics than the standard optimizers in deep learning, and thus have not seen widespread adoption. We find recent deep learning applications of second-order methods (e.g. (Martens & Grosse, 2015; Martens et al., 2018)) to be intriguing, though outside the scope of this paper.

Recently, the role of adaptive regularization has been a hotly contested topic. In (Wilson et al., 2017), the authors suggest that properly-tuned SGD exhibits superior generalization to adaptive methods. In turn, (Keskar & Socher, 2017) propose *switching* the optimizer from Adam to SGD at the end of training, to reap the advantages of each. Influentially, Adam's convergence has been the object of recent scrutiny (Reddi et al., 2018). However, Adam continues to enjoy successful convergence in practice; the problematic construction involves pathological outlier gradients. We do not use the analyses of Adam or AMSGrad.

Several parallel works (Li & Orabona, 2018; Zou & Shen, 2018; Ward et al., 2018; Chen et al., 2018a;b; Zhou et al., 2018) have studied the convergence of adaptive methods for non-convex optimization, matching the asymptotic iteration complexity of SGD. Apart from our algorithmic contribution, our work is (to our knowledge) the first attempt to characterize the *advantage* of adaptivity in terms of the dimension and geometry of the optimization problem.

## 2 THE GGT ALGORITHM

Our main algorithmic contribution is GGT, an efficient first-order algorithm for full-matrix adaptive preconditioning. In brief, GGT uses the preconditioner from full-matrix AdaGrad, with gradient history attenuated exponentially as in Adam, and truncated to a window parameter $r$. The name GGT acts as a convenient mnemonic for the gradient second-moment matrix $\mathbf{G}\mathbf{G}^\top$ maintained by full-matrix AdaGrad, even though *we never compute this matrix*.

Figure 1: Sketch of how GGT performs fast full-matrix preconditioning. Note that the inverse matrices are understood here to be Moore-Penrose pseudoinverses; see Section 2.1 for a full treatment.

The mathematical specification of GGT is given in Algorithm 1, in the usual model of stochastic optimization (see Section 4), with gradients $\widetilde{\nabla} f(x)$. Notice that the coordinate-wise scaling of Adam is recovered by zeroing out the off-diagonal entries of $\mathbf{G}\mathbf{G}^\top$.

---

**Algorithm 1** GGT adaptive optimizer

---

1: **Input:** initializer $x_1$, window size $r$, learning rate schedule $\{\eta_t\}$, $\beta_2 \leq 1, \varepsilon > 0$.
2: **for** $t = 1, \ldots, T$ **do**
3:   Receive stochastic gradient $\widetilde{\nabla} f(x_t)$.
4:   Let $\mathbf{G}_t = [g_t \ g_{t-1} \ \ldots \ g_{t-r+1}]$, where $g_{t-k} := \beta_2^k \widetilde{\nabla} f(x_{t-k})$, or $\mathbf{0}$ if $k \geq t$.
5:   Update $x_{t+1} \leftarrow x_t - \eta_t \cdot [(\mathbf{G}_t \mathbf{G}_t^\top)^{1/2} + \varepsilon \mathbf{I}]^{-1} \widetilde{\nabla} f(x_t)$.
6: **end for**

---

GGT provides the power of full-matrix adaptive regularization at a cost not much larger than SGD. This crucially exploits the fact only a small window of historical gradients are used for preconditioning. The intuition for using a small window, as opposed to the entire history, is clear (and time-tested, by the ubiquity of Adam): the curvature of the loss surface changes, rendering previous gradient information obsolete. We expand on the benefits of forgetting gradients in section 3.1.

The fact that the preconditioning matrix is based on a small window of gradients implies that it has low rank. GGT exploits this fact by computing the inverse square root of the empirical covariance matrix indirectly, as outlined in Figure 1. In effect, instead of inverting a full matrix in the dimension of parameters, using the special matrix structure GGT inverts a matrix of dimension window-size. The remainder of this section will discuss efficient implementation and some heuristics.

GGT has provable guarantees even for non-convex optimization: it is guaranteed to converge to a first-order critical point. Its rate of convergence is never significantly slower than that of SGD, and in some favorable geometric conditions, can be significantly faster. These theoretical bounds are made precise in section 4.

## 2.1 FAST LOW-RANK PRECONDITIONING

The window parameter $r$ should be roughly the number of copies of the model that fit in RAM; in our large-scale experiments, we use $r = 200$. A pessimistic but principled choice is $r = \Theta(1/(1-\beta_2))$, which truncates on the time scale of the exponential attenuation. Our key observation, highlighted in Figure 1, is that the inversion of the large low-rank matrix $\mathbf{G}\mathbf{G}^\top$ can be performed by diagonalizing the small matrix $\mathbf{G}^\top \mathbf{G}$, along with some extremely GPU-friendly matrix-vector operations.

The basic intuition is contained in Figure 1, but it remains to include the $\varepsilon \mathbf{I}$ term. We derive the full update here. Let $\mathbf{G} \in \mathbb{R}^{d \times r}$, $v \in \mathbb{R}^d$ be arbitrary, with $r \leq d$. Write the singular value decomposition $\mathbf{G} = \mathbf{U}\boldsymbol{\Sigma}\mathbf{V}^\top$, with $\mathbf{U} \in \mathbb{R}^{d \times d}, \boldsymbol{\Sigma} \in \mathbb{R}^{d \times r}, \mathbf{V} \in \mathbb{R}^{r \times r}$. Let $\boldsymbol{\Sigma}_d \in \mathbb{R}^{d \times d} := [\boldsymbol{\Sigma} \ 0]$, and let $\boldsymbol{\Sigma}_r \in \mathbb{R}^{r \times r}$ be its top left block. Let $\mathbf{U} =: [\mathbf{U}_r \ \mathbf{U}_{d-r}]$, so that the columns of $\mathbf{U}_r \in \mathbb{R}^{d \times r}$ are an orthonormal basis for the column space of $\mathbf{G}$, and $\mathbf{U}_{d-r} \in \mathbb{R}^{d \times (d-r)}$ its orthogonal component,

noting that $\mathbf{U}_r\mathbf{U}_r^\top + \mathbf{U}_{d-r}\mathbf{U}_{d-r}^\top = \mathbf{I}_d$. Then, we have

$$
\begin{aligned}
\left[(\mathbf{GG}^\top)^{1/2} + \varepsilon\mathbf{I}\right]^{-1}v &= \left[(\mathbf{U}\boldsymbol{\Sigma}_d^2\mathbf{U}^\top)^{1/2} + \varepsilon\mathbf{U}\mathbf{U}^\top\right]^{-1}v = \mathbf{U}(\boldsymbol{\Sigma}_d + \varepsilon\mathbf{I})^{-1}\mathbf{U}^\top v \\
&= \left[\mathbf{U}_r(\boldsymbol{\Sigma}_r + \varepsilon\mathbf{I}_r)^{-1}\mathbf{U}_r^\top + \mathbf{U}_{d-r}(\varepsilon\mathbf{I}_{d-r})^{-1}\mathbf{U}_{d-r}^\top\right]v \\
&= \mathbf{U}_r(\boldsymbol{\Sigma}_r + \varepsilon\mathbf{I}_r)^{-1}\mathbf{U}_r^\top v + \frac{1}{\varepsilon}(\mathbf{I}_d - \mathbf{U}_r\mathbf{U}_r^\top)v \\
&= \frac{1}{\varepsilon}v + \mathbf{U}_r\left[(\boldsymbol{\Sigma}_r + \varepsilon\mathbf{I}_r)^{-1} - \frac{1}{\varepsilon}\mathbf{I}_r\right]\mathbf{U}_r^\top v.
\end{aligned}
$$

The first term is none other than an SGD update step. The rest can be computed by taking the eigendecomposition $\mathbf{G}^\top\mathbf{G} = \mathbf{V}\boldsymbol{\Sigma}_r^2\mathbf{V}^\top$, giving $\mathbf{U}_r = \mathbf{G}\mathbf{V}\sqrt{\boldsymbol{\Sigma}_r}^\dagger$. We prefer this to taking the direct SVD of $\mathbf{G}$, which is $>10$ times slower on GPU.

Using a cyclic buffer to store and update $\mathbf{G}_t$, the algorithm takes $O(dr^2 + r^3)$ (sequential) time per iteration, and $O(dr)$ memory in total. Iterating over the model parameters to update $\mathbf{G}_t$ incurs the same overhead cost as usual adaptive optimizers. The $r \times d$ matrix multiplication and $r \times r$ SVD operations benefit from decades of extensive hardware-level optimizations.

In the experiments in Section 3, we observed a $\sim 1.3\times$ (CNN) and $\sim 2\times$ (RNN) running-time overhead over SGD; we note that this ratio could be even smaller in reinforcement learning (where the environment causes the time bottleneck), or universally with a more optimized implementation.

## 2.2 TWEAKS FOR GGT ON DEEP MODELS

Below, we list some practical suggestions for applying GGT to training large-scale models.

**Momentum.** In order to bring GGT closer to a drop-in replacement for Adam, we can add momentum to the gradient steps: let $v_t \leftarrow \beta_1 v_{t-1} + \widetilde{\nabla}f(x_t)$, and apply the preconditioner to $v_t$ to compute the update step. We use momentum in all large-scale experiments, with the standard $\beta_1 = 0.9$. We also get a small performance boost by using $v_t$ instead of the gradients to update $\mathbf{G}_t$. On the other hand, as long as $r \ll T$, it makes little difference to choose $\beta_2 = 1$, letting the window (rather than exponential attenuation) forget stale gradient information.

**Interpolation with SGD.** We note the possibility of decoupling the scalars $\varepsilon$ and $1/\varepsilon$ which appear in the efficient update step. Appealingly, this allows the user to tune GGT's behavior to be arbitrarily close to that of SGD.

**Numerical concerns.** For greater numerical stability, it is possible to add a small multiple of the identity matrix (we suggest $10^{-6}$) to $\mathbf{G}^\top\mathbf{G}$ before computing its eigendecomposition, without noticeable differences in training.

## 3 EXPERIMENTS

In this section, we present an empirical study of GGT. We begin with some simple experiments, showing that adaptive methods help in the presence of ill-conditioned optimization problems, as well as the value of limited gradient memory. Next, we evaluate the performance of GGT on larger-scale deep learning tasks (and provide some additional such experiments in Appendix B). Finally, we present some interesting empirical insights on the training dynamics in deep learning models. Our visualizations of gradient spectra suggest that adaptive optimizers are indeed correcting for changing anisotropic curvature in the loss landscape.

### 3.1 SYNTHETIC DATA: WHEN DO ADAPTIVITY AND FORGETFULNESS HELP?

The original theorems on the behavior of adaptive first-order methods are established from the perspective of online convex optimization (Duchi et al., 2011). The dynamics are less understood on realistic loss landscapes in stochastic optimization. For this reason, we begin our experimental section with some simple empirical comparisons between full- and diagonal-matrix adaptive optimizers and SGD. Figure 2 summarizes our findings.

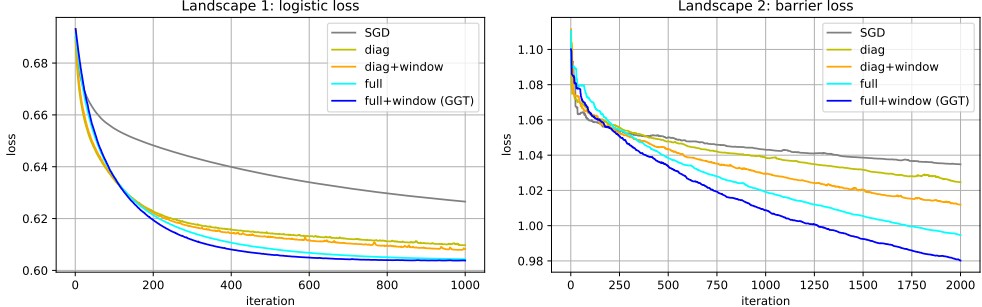

Figure 2: Synthetic experiments on convex loss functions, demonstrating the value of adaptive regularization and attenuation of gradient history. *Left:* An ill-conditioned instance of logistic regression. Adaptive regularization finds a good preconditioner, accelerating optimization. *Right:* Minimizing a barrier function, an example where the curvature changes with position. Optimization is further accelerated by *forgetting* outdated gradient information.

In each synthetic experiment, we generated an ill-conditioned landscape, and compared SGD with adaptive optimizers, excluding the typical accompanying heuristics (i.e. no momentum, regularization, or learning rate schedule). We tested diagonal-matrix preconditioners with and without exponential gradient attenuation (like Adam and AdaGrad, respectively), and their full-matrix analogues. The experiments were robust with respect to the choice of $\varepsilon$ (we used $10^{-4}$) and batch size.

In the first synthetic experiment *(left)*, we exhibit an instance of logistic regression in dimension 10, with $10^3$ samples generated from an extremely anisotropic ($\sigma_{\max}^2/\sigma_{\min}^2 \approx 10^4$) Gaussian distribution, and binary labels determined by a random hyperplane. SGD converges the slowest, and diagonal AdaGrad consistently accelerates optimization. Finally, full-matrix preconditioning (using cubic-time matrix inversion) converges the fastest. In this setting, adding a window improved convergence, but not drastically; we elaborate below.

Next, we show an optimization problem *(right)* which accentuates the utility of exponentially decaying gradient memory. We consider the problem of minimizing the logarithmic barrier function of a randomly generated anisotropic polytope, otherwise known as finding its *analytic center*: this replaces the logistic loss terms with $f_i(w) = -\log(w^\top x_i + c_i)$, with $x_i$ generated the same way as above, and $c_i$ generated uniformly from $[0, 1]$. We observed the same ranking of convergence rates as in the first experiment, but the improvement afforded by the window was much clearer.

The primary conclusion of our synthetic experiments is to demonstrate some small-scale settings in which adaptive regularization ameliorates anisotropy in the optimization landscape. A subtler point is that the windowed variants can help with changing curvature, even for convex losses. Note that the curvature of the former landscape is constant (in that its Hessian matrix at different locations $w$ only changes by a scalar factor). The latter setting, in contrast, features a changing curvature (its Hessians do not commute in general), necessitating "forgetfulness" in adaptive curvature estimation.

In Section 3.4, we will return to these proof-of-concept optimization instances, connecting them to an empirical study of curvature in more realistic landscapes.

## 3.2 GGT ON DEEP CONVOLUTIONAL MODELS

We investigated the training dynamics of GGT on a typical deep architecture for computer vision. For this, we used a 26-layer 3-branch residual network with Shake-Shake regularization, recently proposed in (Gastaldi, 2017). Aside from its ability to reach state-of-the-art classification accuracy, this architecture also features a relatively low parameter count ($\sim 3$M), enabling the use of a large window parameter ($r = 200$).

In each experiment, we kept the cosine learning rate annealing schedule proposed in the paper, originally from (Loshchilov & Hutter, 2016); performance degraded consistently and significantly with a fixed learning rate. For both Adam and GGT, we chose the commonly used parameters $\beta_1 = 0.9, \beta_2 = 0.999, \varepsilon = 10^{-8}$; for SGD, we used momentum with parameter 0.9. With correctly

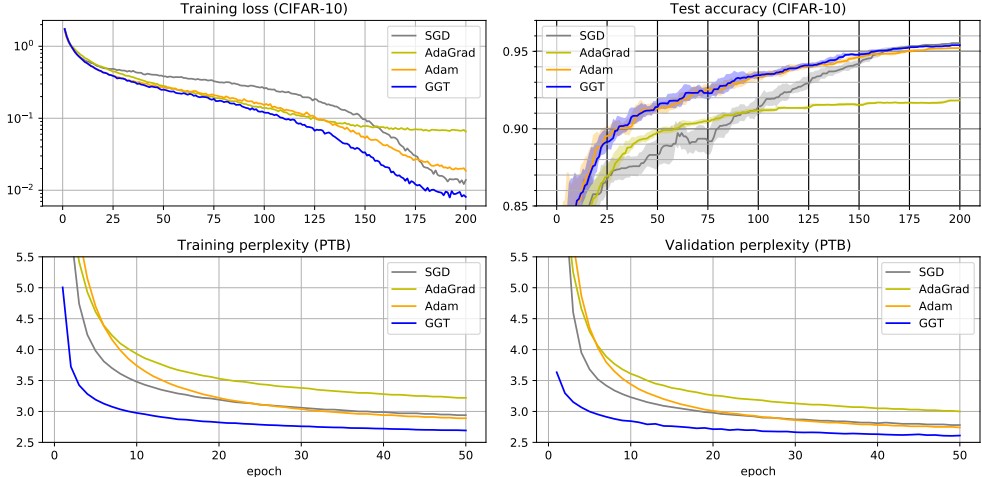

Figure 3: Results of CNN and RNN experiments. GGT dominates in training loss across both tasks, and generalizes better on the RNN task. *Top:* CIFAR-10 classification with a 3-branch ResNet. *Bottom:* PTB character-level language modeling with a 3-layer LSTM.

tuned RMSprop and Adadelta, with the same window parameters, training curves were virtually identical to those for Adam. We used the standard data augmentation techniques of 4-pixel padding + random cropping and horizontal flipping.

Our results are shown in Figure 3 *(top)*. In terms of training loss, GGT consistently dominated existing optimizers. We corroborate a number of observations from previous empirical studies of the generalization of optimizers. Most prominently, we found that SGD generalized slightly better than all others (Wilson et al., 2017; Keskar & Socher, 2017) towards the end of training, including ours. The gap ($< 0.2\%$) is less dramatic than that seen in (Wilson et al., 2017) for two reasons: we only show curves with a tuned and annealed learning rate; also, we use an architecture with powerful explicit regularization techniques which have gained attention since their publication. Our preliminary observation is that GGT shrinks this gap slightly (corroborated by another experiment in Appendix B), and expect that there is vastly more empirical work to be done concerning architectures synergistically tuned to existing optimizers.

We also verify the long-held empirical observation that the learning rate decay of AdaGrad is too aggressive (e.g. in (Zeiler, 2012)), resulting in convergence to a poor solution. Finally, as noted in (Wilson et al., 2017), we find that using a sufficiently low learning rate for any optimizer can result in a better training loss curve, but not without significantly degrading generalization ($> 3\%$ worse).

### 3.3 GGT ON RECURRENT MODELS

Next, we move to recurrent architectures for language modeling. We train a 3-layer LSTM (Hochreiter & Schmidhuber, 1997) with $\sim 5$M parameters for character-level modeling of the Penn Treebank dataset (Marcus et al., 1994). This is the setting in which we observe the most striking improvement over baselines. The particularities of this optimization task, and why it might be especially amenable to full-matrix regularization, remain a fruitful research direction (Pascanu et al., 2013). Figure 3 *(bottom)* shows training and validation perplexities for the first 50 epochs; no optimizer makes significant progress afterwards.

The state of the art for character-level language modeling is less thoroughly documented than its word-level counterpart, though we note that our end-to-end result (validation perplexity $2.42$ after $500$ epochs) is competitive with those reported for recurrent models, like by Krueger et al. (2016). In contrast, Adam, AdaGrad, and SGD reach $2.51$, $2.65$, and $2.76$, respectively. Note that Adam is the *de facto* standard optimizer for language modeling (Melis et al., 2017). Even with iterations taking twice the time, we outperform all baselines in wall-clock time throughout training.

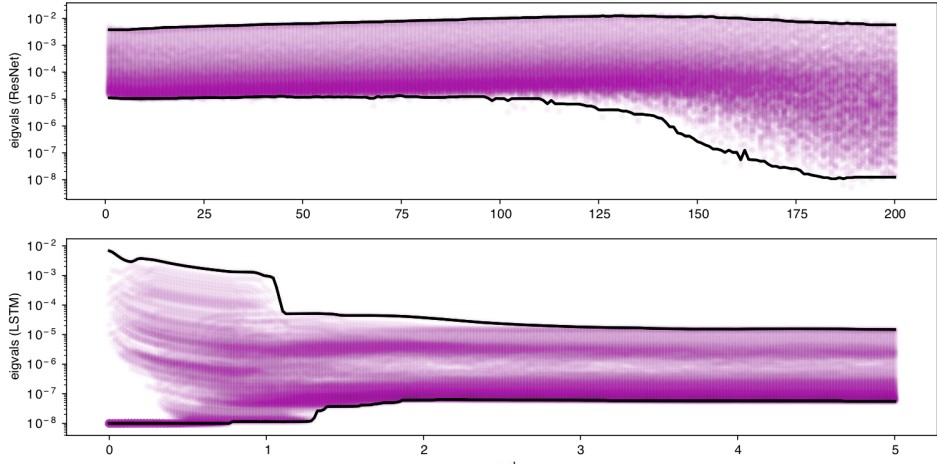

Figure 4: Evolution of the spectrum of the gradient matrix during training. Each vertical slice is a density heatmap of the eigenvalues of $\mathbf{G}_t^\top \mathbf{G}_t$. The black lines indicate the minimum and maximum eigenvalues, smoothed in time by a median filter. *Top:* CNN training. Approaching the end of training, the gradients become more anisotropic. *Bottom:* RNN training. Within the first few epochs, the gradients become more isotropic, then stabilize. (Truncated to 5 epochs; the density was visually stable for the remainder of training.)

We also tried using GGT as a drop-in replacement for Adam in the state-of-the-art word-level language modeling code accompanying (Merity et al., 2017; 2018). Although we were competitive with Adam, we only observed an improvement in the first $\sim 20$ epochs. We hypothesize that the advantage of full-matrix regularization in this setting is more marginal, as the gradients in the embedding layers are naturally sparse in the vocabulary ("one-hot") basis. On a similar note, we found that Adam outperformed GGT on attention-based architectures for NLP; refer to Appendix B for an experiment and discussion.

### 3.4 EMPIRICAL INSIGHTS ON THE SPECTRAL DECAY

In this section, we unify the insights gleaned from the synthetic experiments and deep learning benchmarks. Along the way, we provide some interesting anecdotal observations on the evolution of the preconditioner matrices' singular values.

We plot the density of the spectrum of the low-rank preconditioner $\mathbf{G}_t \mathbf{G}_t^\top$ as training progresses. Since the fast implementation of GGT takes an eigendecomposition of $\mathbf{G}_t^\top \mathbf{G}_t$, we can read off the distribution of eigenvalues during training at no additional computational cost. Figure 4 visualizes the result of this experiment for the CNN and RNN training settings from the previous two sections. In each case, we observe that $\mathbf{G}_t^\top \mathbf{G}_t$ has a condition number of $\sim 10^3$, noting that this can be visualized as the vertical range in the logarithmic plot.

This visualization affords a new way to see how CNN and RNN landscapes are fundamentally different: their gradient spectra evolve in very distinct ways over the course of training. Interestingly, the condition number of the CNN landscape surges near the end, which may be related to the the low-rank structure of well-trained nets noted by Arora et al. (2018), who derive rank-dependent generalization bounds for neural networks. On recurrent models, the rapidly evolving spectral structure at the early stage of training indicates a possibly more complex landscape. Intriguingly, the enormous condition number ($\sim 10^6$) correlates with the massive lead of GGT over the others, confirming our intuition that full-matrix preconditioning ameliorates anisotropy.

To our knowledge, this is the first empirical study of this kind, using the covariance matrix of recent gradients as a surrogate to examining the changing curvature of the loss landscape. In the spirit of recent empirical lenses of this flavor (Raghu et al., 2017; Li et al., 2017), we leave this as a way to visualize deep learning dynamics, possibly of independent exploratory interest.

## 4 A CONVERGENCE RATE ANALYSIS WITH ADAPTIVITY

In this section we outline our analysis of GGT, for which we show convergence to an approximate first-order critical point, in some settings faster than SGD. To obtain the strongest theory, we analyze GGT with a "hard window" instead of exponentially decaying gradient memory, explained in Section A.2.

We work in the usual theoretical framework of stochastic optimization of a differentiable non-convex function $f(\cdot)$, equipped with an unbiased variance-bounded stochastic gradient oracle $\widetilde{\nabla} f(\cdot)$. The objective, as is standard in the literature (see, e.g. Ghadimi & Lan (2013); Allen-Zhu & Hazan (2016)), is to find an $\varepsilon$-approximate stationary point $x$; that is, $\|\nabla f(x)\| \leq \varepsilon$.

### 4.1 THE ADAPTIVE RATIO

We quantify the improvement of adaptive regularization by its advantage over the usual worst-case bound of SGD. To this end, we define the *adaptive ratio* $\mu$ of an algorithm $\mathcal{A}$ as

$$\mu \overset{\text{def}}{=} \frac{f(x_{\mathcal{A}}) - f(x^*)}{\|x_1 - x^*\|_2 \cdot \frac{\sigma}{\sqrt{T}}},$$

where $x_{\mathcal{A}}$ is the output of the $\mathcal{A}$, and $x^*$ is a comparator. For convex optimization problems $x^*$ is naturally the global minimum. For non-convex optimization it is a subtler choice, which we detail in Appendix A.

This ratio for the AdaGrad algorithm was shown in (Duchi et al., 2011) to be always bounded by a quantity independent of $T$, and potentially much smaller. Specifically, it was shown to be inversely proportional to the dimension in certain convex optimization problems, providing a theoretical justification for the speedup of adaptive optimizers. In Section A.4, we show a new, simple, and natural setting illustrating adaptive speedup, even for a *strongly convex* function $f$.

### 4.2 ADAPTIVE CONVERGENCE RATE GUARANTEE

We informally state the main theorem below. We defer the full bound without suppressed smoothness constants, as well as all technical proofs, to Appendix A.

**Theorem 4.1.** *Let $f : \mathbb{R}^d \rightarrow \mathbb{R}$ be a bounded, Lipschitz, and smooth function with stochastic gradient oracle $\widetilde{\nabla} f(\cdot)$, whose variance is at most $\sigma^2$. In expectation, Algorithm 3 outputs an $\varepsilon$-approximate critical point of $f$, with $\widetilde{O}\left(\frac{\mu^2 \sigma^2}{\varepsilon^4}\right)$ calls to $\widetilde{\nabla} f(\cdot)$.*

This theorem matches and potentially improves the known analysis for stochastic gradient descent with the introduction of the data-dependent adaptivity constant $\mu$ into the leading-order term governing the rate of convergence. Since Duchi et al. (2011) bounded $\mu$ by a quantity independent of $T$, our theorem matches the classic $O\left(\varepsilon^{-4}\right)$ rate of convergence.

## 5 CONCLUSION

This work investigates full-matrix adaptive regularization: our main contribution is to make this technique viable for large-scale optimization, by a method for efficient multiplication by the inverse square root of a full second-moment matrix over a short window of gradients. This leads to a new algorithm, GGT, a truly scalable optimization algorithm with full-matrix adaptive preconditioning.

Through synthetic experiments, we have shown that GGT accelerates optimization in ill-conditioned loss landscapes; this is supported by accompanying adaptive convergence guarantees. Preliminary experiments show accelerated convergence on standard deep learning benchmarks, with very different training dynamics from existing diagonal adaptive methods. We accompany our algorithm and experiments with the first theoretical characterization of the benefits of adaptive regularization in a non-convex setting. We hope that GGT will be the first of a new class of algorithms for the modern large-scale optimization toolbox, and to foster new discussion towards an ever-elusive understanding of loss landscapes in deep learning.

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

## A  FULL ADAPTIVE CONVERGENCE ANALYSIS

In this section, we give the details on the theoretical treatment of GGT outlined in Section 4. The overall goal is to develop a theory for adaptive regularization in non-convex stochastic optimization. After formalizing the setting, we will define a version of GGT that uses a hard gradient memory window. This will allow us to transfer any insight on the advantage of adaptivity in the convex case to the non-convex case, giving rise to the main theorem. We will conclude this section by with an example illustrating the advantage of adaptive optimizers in the presence of sparse gradients.

### A.1  SETTING: STOCHASTIC NON-CONVEX OPTIMIZATION

Theorem A.2 will provide a bound on the number of stochastic gradient calls required by GGT to achieve a first-order critical point. In particular, the theorem shows that GGT can converge to an approximate first-order critical point faster than SGD, with convergence rate controlled by the *adaptive ratio* $\mu$, defined in (1).

We consider the standard setting of stochastic optimization of a differentiable non-convex function $f(\cdot)$, equipped with a bounded-variance stochastic gradient oracle defined as follows.

**Definition A.1** (stochastic gradient oracle). *Given a function $f : D \to \mathbb{R}$ we call an oracle $O_f$, a $\sigma$-bounded stochastic gradient oracle if for any $x$, $O_f$ returns a a random vector $\widetilde{\nabla} f(x)$ such that*

$$\mathbb{E}\left[\widetilde{\nabla} f(x)\right] = \nabla f(x) \quad and \quad \mathbb{E}\left[\|\widetilde{\nabla} f(x) - \nabla f(x)\|^2\right] \leq \sigma^2.$$

The objective, as is standard in non-convex optimization, is to find a first-order critical point, i.e. a point $x$ for which $\|\nabla f(x)\| \leq \varepsilon$. We will also assume that $f$ has a Lipschitz gradient; i.e. $\|\nabla^2 f(x)\|_2 \leq L$.

Our algorithm makes a reduction to the case of stochastic convex optimization. The setting formally is that, given a smooth convex function and a $\sigma$-bounded stochastic gradient oracle, the algorithm's aim is to minimize the convex function $f$. Given any algorithm $\mathcal{A}$ we can now define the *adaptive ratio* of the algorithm, referred to as $\mu$, as

$$\mu \overset{\text{def}}{=} \frac{f(x_{\mathcal{A}}) - f(x^*)}{\|x_1 - x^*\|_2 \cdot \frac{\sigma}{\sqrt{T}}} \tag{1}$$

where $x_{\mathcal{A}}$ is the output of the algorithm $\mathcal{A}$ and $x^* \in \text{argmin}_x f(x)$, with a total of at most $T$ calls to the stochastic gradient oracle. $\mu$ captures the advantage in convergence rate obtained by the algorithm as compared to the error obtained by vanilla SGD, noting that the denominator is a bound on the error obtained by SGD in the same setting.

A popular algorithm for stochastic (and in general online) convex optimization is AdaGrad (Duchi et al., 2011). Due to adaptive regularization, AdaGrad can often be advantageous over SGD. We quantify this advantage by the notion of $\mu$ defined above. The bounds of Duchi et al. (2011) imply that $\mu$ can be as small as $\frac{1}{\sqrt{d}}$, depending on the geometry of the optimization problem. An example of this was provided by Duchi et al. (2011) for both the diagonal and the full version of Adagrad. At the end of this section, we provide a different example which shows the same phenomenon even in the case of strongly convex functions.

In the rest of this section we describe Algorithm 3, which uses AdaGrad (Algorithm 2) as a subroutine during each window. In this regard, while stating the bounds for our algorithms, we use $\mu$ as an upper bound on the advantage of AdaGrad in each iteration.

### A.2  A SUITABLE ABSTRACTION FOR GGT

As mentioned in Section 4, our analysis uses a slightly idealized version of GGT, which replaces the gradient memory mechanism (governed by $w$ and $\beta_2$) with a *hard* window; i.e., the gradient buffer is *reset* every $w$ steps. This simple modification enables us to develop a more informative theory, in which we benefit directly from the familiar theory of AdaGrad for convex optimization, while capturing the necessity of forgetting past gradient information in adaptive non-convex optimization.

First, for clarity, we restate the definition of the full-matrix AdaGrad algorithm, introduced by Duchi et al. (2011), which accumulates the second-moment matrix of all past gradients:

---

**Algorithm 2** AdaGrad for convex optimization (Duchi et al., 2011)

---

1: **Input:** initializer $x_1$, window length $w$, stochastic gradient oracle $\widetilde{\nabla} f(\cdot)$, $\varepsilon, \eta > 0$.
2: **for** $t = 1, \ldots, w$ **do**
3:     Receive stochastic gradient $\widetilde{\nabla} f(x_t)$.
4:     Let $\mathbf{G}_t = [g_t \ g_{t-1} \ldots g_1]$, where $g_t := \widetilde{\nabla} f(x_t)$.
5:     Update $x_{t+1} \leftarrow x_t - \eta \cdot \left[\varepsilon \mathbf{I} + (\mathbf{G}_t \mathbf{G}_t^\top)^{1/2}\right]^{-1} g_t$.
6: **end for**
7: **Output:** Average iterate $\frac{1}{w}\left(\sum_{t=1}^{w} x_t\right)$.

---

The final algorithm we analyze simply runs AdaGrad between restarts.

---

**Algorithm 3** GGT with a hard gradient window

---

1: **Input:** initializer $x_1$, time horizon $T$, window length $w$, $\lambda > 0$.
2: **for** $t = 1$ to $T$: **do**
3:     Let $f_t(x) = f(x) + \lambda \|x - x_t\|^2$.
4:     Update $x_{t+1}$ to be the output of Algorithm 2 on $f_t(x)$, starting at $x_t$, for $w$ steps.
5: **end for**
6: **Output:** Best iterate $x_{t^*}$, where $t^* := \operatorname{argmin}_{t \leq T+1} \|\nabla f(x_t)\|$.

---

The remaining discrepancies between Algorithm 3 and Algorithm 1 from the main paper are standard. We provide some references below.

- **Absence of first-moment estimation.** Although it is customary to use nonzero $\beta_1$ (otherwise known as momentum) when applying Adam in practice, it is orthogonal to the effect of adaptive regularization in all established theory. In fact, the convergence rates given by Kingma & Ba (2014) (and fixed by Reddi et al. (2018)) contain only factors of $1/(1 - \beta_1)$, and are thus strongest when $\beta_1 = 0$.

- **Model averaging.** Theoretical guarantees in online and stochastic convex optimization are most naturally stated on the average iterate; see (Polyak & Juditsky, 1992; Duchi et al., 2011). Thus, we adopt the convention that Algorithm 2 returns the average iterate. We note that model averaging is a common regularization technique in practical non-convex settings, though not the default choice for adaptive optimizers in practice.

- $\ell_2$ **regularization.** The addition of the $\lambda \|x - x_t\|^2$ term in Algorithm 3 is an artifact we introduce to obtain a tight analysis for hard-window GGT. It ensures that iterates in each window do not move too far, and allows us to analyze each window as a fixed convex program, so that we can use the convex theory of AdaGrad directly. The soft-window analogue would simply to be decrease the learning rate. Interestingly, a similar technique directly appears in the algorithm proposed by Allen-Zhu (2017). Finally, we note that from a $\sigma$-bounded stochastic gradient oracle for $f$, it is trivial to construct one for $f_t$, by adding $-2\lambda x_t$ (deterministically).

## A.3 MAIN THEOREM AND PROOF

**Theorem A.2.** *Consider a non-convex function $f$, such that for all $x$, $\|\nabla^2 f(x)\|_2 \leq L$ and a point $x_1$ such that $f(x_1) - \min_{x^* \in \mathcal{K}} f(x^*) \leq M$. Further, suppose we have access to a $\sigma$-bounded stochastic gradient oracle $O_f$. Suppose for any $\lambda \geq \frac{L}{2}$, Algorithm 3 is run with $T = \frac{4M(L+2\lambda)}{\varepsilon^2}$ and $w = \frac{16\mu^2 \sigma^2 (L+2\lambda)}{\varepsilon^2 (2\lambda - L)}$. Then the point $x'$ returned by Algorithm 3 is such that*

$$\mathbb{E}\|\nabla f(x')\| \leq \varepsilon,$$

*where $\mu = \max_{t \in [T]} \mu_t$ and $\mu_t$ is the adaptive ratio when run on $f_t$ (as defined in (1)). Further, note that choosing $\lambda = 3L/2$, the total number of stochastic gradient calls to the oracle $O_f$, made by the algorithm is bounded by $T \cdot w = \frac{512 L M \mu^2 \sigma^2}{\varepsilon^4}$.*

For the setting of Theorem A.2, the best known bound on the number of oracle calls to the stochastic gradient oracle in the case of the vanilla SGD algorithm is $O(\frac{LM\sigma^2}{\varepsilon^4})$. Note that due to the presence of $\mu^2$ in the bound provided in Theorem A.2 reflects the advantage of Algorithm 3 over SGD. This advantage as we argue in the following section can be as large as up to a factor of $1/d$, a significant improvement over SGD.

Before proving Theorem A.2, we state an oracle complexity bound for AdaGrad (Algorithm 2) for strongly convex functions.

**Lemma A.3.** *Suppose f is a $\lambda$-strongly convex function equipped with a $\sigma$-bounded stochastic gradient oracle. Given an initial point $x_1$, Algorithm 2 when run for $w$ steps is guaranteed to output a point $x'$ such that*

$$\mathbb{E}[f(x')] - \min_x f(x) \leq \frac{\mu^2\sigma^2\sqrt{2(f(x_1) - \min_x f(x))}}{\sqrt{\lambda w}},$$

*where $\mu$ is the adaptive ratio of AdaGrad on $f$ as defined in* (1).

Using this lemma we first prove Theorem A.2 and then finish the section by providing a proof of Lemma A.3.

*Proof of Theorem A.2.* We begin by proving the following useful property regarding the function $f_t$ for any $t$ and any $\eta$:

$$f_t(x_t) - \min_x f_t(x) \geq f(x_t) - f_t(x_t - \eta\nabla f(x_t))$$

$$= f(x_t) - f(x_t - \eta\nabla f(x_t)) - \lambda\eta^2\|\nabla f(x_t)\|^2$$

$$\geq \eta\|\nabla f(x_t)\|^2 - \frac{L\eta^2}{2}\|\nabla f(x_t)\|^2 - \lambda\eta^2\|\nabla f(x_t)\|^2.$$

Setting $\eta = \frac{1}{L+2\lambda}$, we get that

$$f_t(x_t) - \min_x f_t(x) \geq \frac{\|\nabla f(x_t)\|^2}{2(L+2\lambda)}. \tag{2}$$

We will now prove the theorem by contradiction. Suppose for all the $t$, $\|\nabla f(x_t)\|^2 > \varepsilon^2$. We now have that

$$f(x_t) - f(x_{t+1}) \geq f_t(x_t) - f_t(x_{t+1})$$
$$= f_t(x_t) - \min_x f_t(x) - (f_t(x_{t+1}) - \min_x f_t(x))$$
$$\geq f_t(x_t) - \min_x f_t(x) - \frac{\sqrt{f_t(x_t) - \min_x f_t(x)}\sqrt{\varepsilon^2}}{2\sqrt{2(L+2\lambda)}}$$
$$\geq f_t(x_t) - \min_x f_t(x) - \frac{\sqrt{f_t(x_t) - \min_x f_t(x)}\sqrt{\|\nabla f(x_t)\|^2}}{2\sqrt{2(L+2\lambda)}}$$
$$\geq \frac{f_t(x_t) - \min_x f_t(x)}{2} \geq \frac{\|\nabla f_t(x_t)\|^2}{4(L+2\lambda)} > \frac{\varepsilon^2}{4(L+2\lambda)}. \tag{3}$$

where the first inequality follows from noting that $f(x) \leq f_t(x)$ for all $x \in \mathbb{R}^d$, and that $f_t(x_t) = f(x_t)$. The second inequality follows from Lemma A.3, by noting that $f_t$ is $2\lambda - L$ strongly convex and the choice of $w$. The third inequality follows from the counterfactual assumption $\|\nabla f(x_t)\|^2 > \varepsilon^2$, and the last set of inequalities follow from (2).

Summing (3) over all $t \in [T]$ gives us that

$$f(x_1) - f(x_{T+1}) > \frac{T\varepsilon^2}{4(L+2\lambda)} = M,$$

which is a contradiction and hence proves the theorem. The number of stochastic gradient oracle calls when $\lambda = 3L/2$ is bounded by

$$T \cdot w \leq \frac{4M(L+2\lambda)}{\varepsilon^2} \cdot \frac{16\mu^2\sigma^2(L+2\lambda)}{\varepsilon^2(2\lambda - L)} \leq \frac{512M\mu^2\sigma^2L}{\varepsilon^4}.$$

$\square$

*Proof of Lemma A.3.* We have

$$\mathbb{E}\left[f(x')\right] - f(x^*) = \mathbb{E}\left[\frac{\mu\sigma}{\sqrt{w}}\|x_1 - x^*\|\right] \leq \mathbb{E}\left[\frac{\mu\sigma\sqrt{2(f(x_1) - f(x^*))}}{\sqrt{w\lambda}}\right],$$

where the equality follows from the definition of $\mu$ and the inequality follows from strong convexity. $\square$

### A.4   EXAMPLE: THE ADVANTAGE OF ADAPTIVITY

Here we provide a strongly convex function (in fact a simple quadratic) and a sketch of the proof of the fact that depending on the starting point adaptive advantage i.e. $\mu$ of AdaGrad can be up to a factor of $\sqrt{d}$.

Consider the function $\|x\|^2$ in $\mathbb{R}^d$ and consider the starting point $x_0$. Let the stochastic gradient oracle $O_f$ be such that before the experiment the oracle samples a random orthonormal basis $V = \{v_1 \dots v_d\}$ and when queried at a point $x$ returns the vector

$$\widetilde{\nabla} f(x) = \nabla f(x) + a_t z_t$$

where $a_t = \pm 1$ with probability $1/2$ and $z_t$ is a vector picked from the set $V$ uniformly randomly. It is easy to verify that $O_f$ is a $\sigma$-bounded stochastic gradient oracle. We now provide an analysis of AdaGrad with the above oracle for $f$.

Firstly note that we can without loss of generality, assume that the basis chosen is the canonical basis $\{e_i\}$. This can be seen by performing a simple rotation which does not affect the function $\|x\|^2$. Further under this setting note that AdaGrad is equivalent to running a one dimensional SGD algorithm in each coordinate independently. The following bound now follows directly from the well known analysis of SGD on smooth functions (see Theorem 6.3 in Bubeck et al. (2015) for a concrete reference).

$$\forall \ i \in [d] \quad (x'[i])^2 \lesssim \frac{|x_1[i]|\sqrt{\sum_{t=1}^T (\sigma_t[i])^2}}{T} = |x_1[i]| \cdot \sqrt{\frac{1}{Td}},$$

where $\sigma_t[i] = 1/d$ is the variance of the noise in the stochastic gradient seen at time $t$ along coordinate $i$ and $x'$ is the output of AdaGrad. Note that in the above we have ignored the *bias* term which scales as $1/T$ (refer to Theorem 6.3 in Bubeck et al. (2015)). This implies that the overall error for AdaGrad scales as

$$\|x'\|^2 \lesssim \|x_1\|_1 \cdot \sqrt{\frac{1}{Td}}.$$

Therefore the advantage of adaptivity $\mu$ is bounded by

$$\mu \leq \frac{\|x_1\|_1 \sqrt{\frac{1}{Td}}}{\|x_1\|_2 \sqrt{\frac{1}{T}}} = \frac{\|x_1\|_1}{\|x_1\|_2 \sqrt{d}}.$$

This follows by noting that the variance of the noise in the stochastic gradient measured in the $\ell_2$ is 1. The above expression implies that $\mu$ can be as small as $O(\frac{1}{\sqrt{d}})$ in particular if the starting point $x_1$ is sparse or nearly sparse and therefore $\|x_1\|_1 \sim \|x_1\|_2$.

## B   ADDITIONAL EXPERIMENTS

### B.1   COMPARISON OF WALL CLOCK TIME

For those interested in end-to-end performance in terms of model training speed, we provide an alternate visualization for the experiments in Sections 3.2 and 3.3, replacing the epoch count with total cumulative training time on the horizontal axis. Evidently, on the LSTM task, GGT outperforms the baselines at all times (and converges upon a better solution), even with the additional time overhead.

The same categorical improvement was not observed on the vision task, for training convergence or generalization. This is probably due to the interactions between modern convolutional architectures and the epoch-dependent learning rate schedule, which we have not attempted to re-tune.

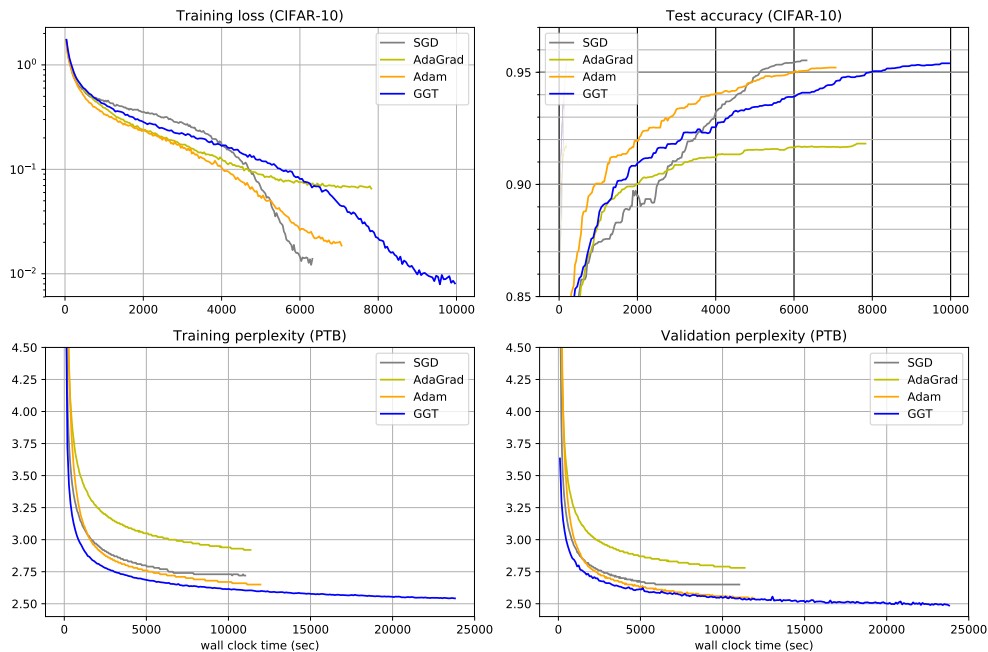

Figure 5: Plot of experiments from Sections 3.2 and 3.3, with wall clock time on horizontal axis instead of epoch count (as in Figure 3). *Top:* CIFAR-10 classification with a 3-branch ResNet. *Bottom:* PTB character-level language modeling with a 3-layer LSTM.

### B.2 EXPERIMENTS ON ADDITIONAL ARCHITECTURES

We present some additional large-scale empirical studies in Figure 6.

To demonstrate a vision task with a harder optimization landscape, we use GGT to train a 19-layer "vanilla" convolutional network (VGGNet, Simonyan & Zisserman (2014)), without residual connections or batch normalization, on the same CIFAR-10 classification task. Here, we recover the same insights as found by Wilson et al. (2017), in which diagonal-matrix adaptive methods can fail to train a network dramatically. Here, unlike diagonal-matrix adaptive optimizers, GGT stays on par with SGD throughout training, with a $\sim 1\%$ gap remaining in generalization at the end. We use a standard fixed halving learning rate schedule; it is clear here that in the initial epochs after decaying the learning rate, GGT trains the most rapidly. We leave a careful investigation of leveraging this phenomenon, and tuning GGT's learning rate schedule, to future work.

A recent significant advancement on many NLP tasks, including language modeling, is the introduction of attention-based models. We investigate the behavior of GGT on a Transformer network (Vaswani et al., 2017), on the same Penn Treebank character-level language modeling task. Here, after an initial lead, GGT is outperformed by Adam in training and validation loss. The value of using gradient correlations to assist in the training of attention models seems to be limited.

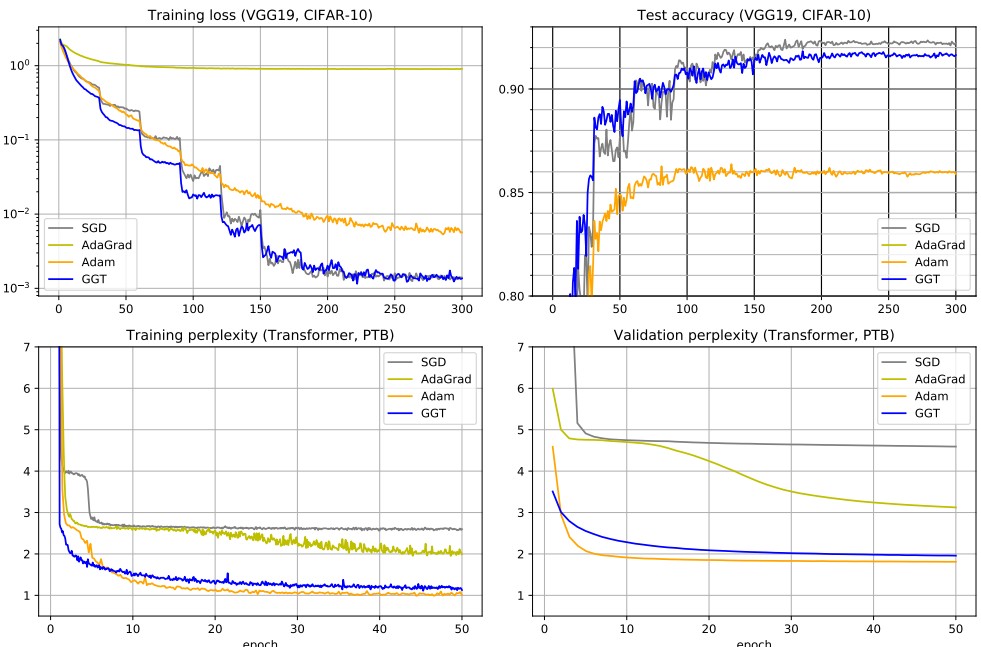

Figure 6: Plot of experiments from Sections 3.2 and 3.3, with wall clock time on horizontal axis instead of epoch count (as in Figure 3). *Top:* CIFAR-10 classification with a 19-layer vanilla CNN. *Bottom:* PTB character-level language modeling a Transformer network.

