# OpenReview forum: "The Case for Full-Matrix Adaptive Regularization"
_ICLR.cc/2019/Conference_

### Official Review · AnonReviewer2 · 2018-11-02
**see review**

**Rating:** 5
**Confidence:** 3

**Review:**

The paper considers adaptive regularization, which has been popular in neural network learning.  Rather than adapting diagonal elements of the adaptivity matrix, the paper proposes to consider a low-rank approximation to the Gram/correlation matrix.

When you say that full-matrix computation "requires taking the inverse square root", I assume you know that is not really correct?  As a matter of good implementation, one never takes the inverse of anything.  Instead, on solves a linear system, via other means.  Of course, approximate linear system solvers then permit a wide tradeoff space to speed things up.

There are several issues convolved here: one is ``full-matrix,'' another is that this is really a low-rank approximation to a matrix and so not full matrix, another is that this may or may not be implementable on GPUs.  The latter may be important in practice, but it is orthogonal to the full matrix theory.

There is a great deal of discussion about full-matrix preconditioning, but there is no full matrix here.  Instead, it is a low-rank approximation to the full matrix.  If there were theory to be had here, then I would guess that the low-rank approximation may work even when full matrix did not, e.g., since the full matrix case would involve too may parameters.

The discussion of convergence to first order critical points is straightforward.

Adaptivity ratio is mentioned in the intro but not defined there.  Why mention it here, if it's not being defined.

You say that second order methods are outside the scope, but you say that your method is particularly relevant for ill-conditioned problems.  It would help to clarify the connection between the Gram/correlation matrix of gradients and the Hessian and what is being done to ill-conditioning, since second order methods are basically designed for ill-conditioned problems..

It is difficult to know what the theory says about the empirical results, given the tweaks discussed in Sec 2.2, and so it is difficult to know what is the benefit of the method versus the tweaks.

The results shown in Figure 4 are much more interesting than the usual training curves which are shown in the other figures.  If this method is to be useful, understanding how these spectral properties change during training for different types of networks is essential.  More papers should present this, and those that do should do it more systematically.

You say that you "informally state the main theorem."  The level of formality/informality makes it hard to know what is really being said.  You should remove it if it is not worth stating precisely, or state it precisely.  (It's fair to modularize the proof, but as it is it's hard to know what it's saying, except that your method comes with some guarantee that isn't stated.)

---

> ### Author Response · Authors · 2018-11-20
> **Response to AnonReviewer2**
>
> Thanks for the review.
>
> There are two significant inaccuracies:
> 1. GGT does not take the view of a low-rank *approximation*. This is a central point of the paper.
> 2. Re: iterative methods: the preconditioner is a -1/2 power of the Gram matrix, not the inverse.
>
> More details below:
>
> @Inverse square root: We are fully aware of the distinction.
> - Note that iterative solvers like conjugate gradient do not immediately apply here, as we are solving a linear system in M^{1/2}, not M.
> - Krylov subspace iterative solvers suffer from a condition number dependence, incurring a hard tradeoff between iteration complexity and \eps. [1]
> - We actually *did* try polynomial approximations to M^{-1/2} as an alternative to our proposed small-SVD step. We saw worse approximation (the condition number dependence kicks in) and worse GPU performance (parallel computation time scales with polynomial degree).
>
> @Full-matrix terminology: The use of “full-matrix” to distinguish from “diagonal-matrix” is standard, and taken directly from [2].
>
> @Full-matrix vs. full-rank: Note that we do not consider the windowed Gram matrix to be an “approximation” of the “full” gram matrix. The window is for the purpose of forgetting gradients from the distant past, motivated by (1) our theory, (2) the small-scale synthetic experiments, and (3) the extreme ubiquity of Adam and RMSprop, which do the same. Note that we do no approximation on the windowed Gram matrix, the fact that it is low rank is a feature.
>
> @Location of \mu definition: Is the reviewer’s suggestion simply to move this definition into the intro?
>
> @Comparison with second-order methods: Please refer to our response to Reviewer 1 for some additional comments.
>
> @Tweaks: We don’t believe that any of the tweaks should be so controversial.
> - The \eps parameters are present in *every* adaptive optimizer, for stability. The interpolation with SGD is just another take on this.
> - The exponential smoothing of the first moment estimator is a subtler point. As we point out in Appendix A.2, in the theory for Adam/AMSgrad [3,4], \beta_1 *degrades* the moment estimation, yet everyone uses momentum in practice. Even if this is unconvincing, the performance gap upon removing this tweak is minor, and our empirical results hold without this tweak. We are simply offering a heuristic that we have observed to help training unconditionally, just like momentum in Adam.
>
> @Informal main theorem: By “informal” we truly mean that we are suppressing the smoothness constants (L, M) for readability and space constraints. We are simply adopting the widespread practice of deferring the non-asymptotic mathematical statement to the appendix.
>
> [1] Tight complexity bounds for optimizing composite objectives. Blake E Woodworth, Nati Srebro. NIPS 2016.
> [2] Adaptive subgradient methods for online learning and stochastic optimization. J Duchi, E Hazan, Y Singer. JMLR 2012.
> [3] Adam: A Method for Stochastic Optimization. D.Kingma,J. Ba. ICLR 2015.
> [4] On the Convergence of Adam and Beyond. S. Reddi, S. Kale, S. Kumar. ICLR 2018.

---

### Official Review · AnonReviewer1 · 2018-11-02
**Elegant idea, but the I'm not convinced that the benefits outweigh the increased computational cost**

**Rating:** 5
**Confidence:** 3

**Review:**

The authors seek to make it practical to use the full-matrix version of Adagrad’s adaptive preconditioner (usually one uses the diagonal version), by storing the r most recently-seen gradient vectors in a matrix G, and then showing that (GG^T)^(-½) can be calculated fairly efficiently (at the cost of one r*r matrix inversion, and two matrix multiplications by an r*d matrix).

This is a really nice trick. I’m glad to see that the authors considered adding momentum (to adapt ADAM to this setting), and their experiments show a convincing benefit in terms of performance *per iteration*. Interestingly, they also show that the models found by their method also don’t generalize poorly, which is noteworthy and slightly surprising.

However, their algorithm--while much less computationally expensive than true full-matrix adaptive preconditioning---is still far more expensive than the usual diagonal version. In Appendix B.1, they report mixed results in terms of wall-clock time, and I strongly feel that these results should be in the main body of the paper. One would *expect* the proposed approach to work better than diagonal preconditioning on a per-iteration basis (at least in terms of training loss). A reader’s most natural question is whether there is a large enough improvement to offset the extra computational cost, so the fact that wall-clock times are relegated to the appendix is a significant weakness.

Finally, the proposed approach seems to sort of straddle the line between traditional convex optimization algorithms, and the fast stochastic algorithms favored in machine learning. In particular, I think that the proposed algorithm has a more-than-superficial resemblance to stochastic LBFGS: the main difference is that LBFGS approximates the inverse Hessian, instead of (GG^T)^(-½). It would be interesting to see how these two algorithms stack up.

Overall, I think that this is an elegant idea and I’m convinced that it’s a good algorithm, at least on a per-iteration basis. However, it trades-off computational cost for progress-per-iteration, so I think that an explicit analysis of this trade-off (beyond what’s in Appendix B.1) must be in the main body of the paper.

---

> ### Author Response · Authors · 2018-11-20
> **Response to AnonReviewer1**
>
> Thanks for the review.
>
> @Wall-clock: We don’t quite understand the question. As mentioned in the response to Reviewer 3, our NLP example does answer the natural question about end-to-end gains. Is the reviewer only concerned with the location of the plots?
> - Another note: to perform a full wall-clock comparison with algorithms that have different per-iteration costs, one must disentangle and retune various hyperparameter choices, most notably the learning rate schedule. Thus we decided to feature the per-iteration comparison in the main paper, as it is the cleanest one.
>
> @L-BFGS: On a high level, we agree that GGT develops a similar window-based approximation to the gradient Gram matrix as L-BFGS does to the approximated Hessian. While adaptive methods have proven effective in practice, quasi-Newton algorithms are not in general regarded as competitive for deep learning (despite recent efforts [1,2]), and that’s why it is not compared to in the vast majority of deep learning papers.
> - Quasi-Newton methods are suited for deterministic problems, while stochasticity is crucial in deep learning. This is because they try to approximate the Hessian by finite differences, which seems unstable with stochastic gradients in practice.
> - Direct second-order methods require significant modifications to converge in the non-convex setting (see [3,4]). Even these have not been observed to work well in deep learning.
> - One reason for the practical success of AdaGrad-like algorithms we believe is the difference of  -1/2 vs. -1 power on the Gram matrix, which seems to change the training dynamics dramatically. With the gradient Gram matrix and a -1 power, meaningful end-to-end advances have only been claimed for niche tasks other than classification.
>
> [1] Stochastic L-BFGS: Improved Convergence Rates and Practical Acceleration Strategies. R. Zhao and W. Haskell and V. Tan. arXiv, 2017.
> [2] A Stochastic Quasi-Newton Method for Large-Scale Optimization. R. Byrd, S. Hansen, and J. Nocedal, and Y. Singer SIAM Journal on Optimization, 2016.
> [3] Accelerated methods for nonconvex optimization. Y. Carmon, J. Duchi, O. Hinder, A. Sidford. SIAM Journal on Optimization, 2018.
> [4] Finding approximate local minima faster than gradient descent. N. Agarwal, Z. Allen-Zhu, B. Bullins, E. Hazan, and T. Ma. STOC 2017.

---

### Official Review · AnonReviewer3 · 2018-11-03
**How to make sgd with full matrix pre-conditioning scalable?**

**Rating:** 6
**Confidence:** 3

**Review:**

adaptive versions of sgd are commonly used in machine learning. adagrad, adadelta are both popular adaptive variations of sgd. These algorithms can be seen as preconditioned versions of gradient descent where the preconditioner applied is a matrix of second-order moments of the gradients. However, because this matrix turns out to be a pxp matrix where p is the number of parameters in the model, maintaining and performing linear algebra with this pxp matrix is computationally intensive. In this paper, the authors show how to maintain and update this pxp matrix by storing only smaller matrices of size pxr and rxr, and performing 1. an SVD of a small matrix of size rxr 2. matrix-vector multiplication between a pxr matrix and rx1 vector. Given that rxr is a small constant sized matrix and that matrix-vector multiplication can be efficiently computed on GPUs, this matrix adapted SGD can be made scalable. The authors also discuss how to adapt the proposed algorithm with Adam style updates that incorporate momentum. Experiments are shown on various architectures (CNN, RNN) and comparisons are made against SGD, ADAM.

General comments: THe appendix has some good discussion and it would be great if some of that discussion was moved to the main paper.

Pros:  Shows how to make full matrix preconditioning efficient, via the use of clever linear algebra, and GPU computations.
Shows improvements on LSTM tasks, and is comparable with SGD, matching accuracy with time.

Cons: While doing this leads to better convergence, each update is still very expensive compared to standard SGD, and for instance on vision tasks the algorithm needs to run for almost double the time to get similar accuracies as an SGD, adam solver.  This means that it is not apriori clear if using this solver instead of standard SGD, ADAM is any good. It might be possible that if one performs few steps of GGT optimizer in the initial stages and then switches to SGD/ADAM in the later stages, then some of the computational concerns that arise are eliminated. Have the authors tried out such techniques?

---

> ### Author Response · Authors · 2018-11-20
> **Response to AnonReviewer3**
>
> Thanks for the review.
>
> @Update overhead: We argue that per-iteration performance is a worthwhile objective in itself, which is less significant in some scenarios (e.g. costly function evaluation, like in RL, or expensive backprops, like in RNNs). That said, we were indeed not able to demonstrate end-to-end gains in vision. Please note that in the NLP benchmark our algorithm finds a better solution and wins in wall-clock time.
>
> @Switching: This is a good suggestion, and we indeed do cite one of the papers attempting to approach optimizer-switching in a principled way. We found that we could squeeze out some wall-clock gains by applying the expensive update more sparingly, but the value of including this in the paper was unclear (effectively adding a host of hyperparameters orthogonal to the central idea).

---

### Meta-Review · Area_Chair1 · 2018-12-13
**Needs more focus on wall clock time and more analysis of the relationship to similar approaches**

**Confidence:** 5
**Recommendation:** Reject

**Metareview:**

This paper shows how to implement a low-rank version of the Adagrad preconditioner in a GPU-friendly manner. A theoretical analysis of a "hard-window" version of the proposed algorithm demonstrates that it is not worse than SGD at finding a first-order stationary point in the nonconvex setting. Experiments on CIFAR-10 classification using a ConvNet and Penn Treebank character-level language modeling using an LSTM show that the proposed algorithm improves training loss faster than SGD, Adagrad, and Adam (measuring time in epochs) and has better generalization performance on the language modeling task. However, if wall-clock time is used to measure time, there is no speedup for the ConvNet model, but there is for the recurrent model. The reviewers liked the simplicity of the approach and greatly appreciated the elegant visualization of the eigenspectrum in Figure 4. But, even after discussion, critical concerns remained about the need for more focus on the practical tradeoffs between per-iteration improvement and per-second improvement in the loss and the need for a more careful analysis of the relationship of this method to stochastic L-BFGS. A more minor concern is that the term "full-matrix regularization" seems somewhat deceptive when the actual regularization is low rank. The AC also suggests that, if the authors plan to revise this paper and submit it to another venue, they consider the relationship between GGT and the various stochastic natural gradient optimization algorithms in the literature that differ from GGT primarily in the exponent on the Gram matrix.